# Effect of Surface Treatment and Storage Time on Immediate Repair Bond Strength Durability of Methacrylate- and Ormocer-Based Bulk Fill Resin Composites

**Farid S. El-Askary** [1] , **Sara A. Botros** [2,*] **and Mutlu Özcan** [3]

[1]  Operative Dentistry Department, Faculty of Dentistry, Ain Shams University, Cairo 11566, Egypt; faridelaskary@asfd.asu.edu.eg

[2]  Restorative Dentistry Department, Faculty of Dentistry, The British University in Egypt, El Sherouk City, Cairo 11837, Egypt

[3]  Center of Dental Medicine, Division of Dental Biomaterials, Clinic for Reconstructive Dentistry, University of Zürich, 8006 Zürich, Switzerland; mutlu.ozcan@zzm.uzh.ch

*  Correspondence: sarah.botros@bue.edu.eg; Tel.: +20-10-0038-1136

**Abstract:** The aim of this study was to evaluate the effect of surface treatment and storage time on immediate repair bond durability of methacrylate- and ormocer-based bulk fill composites. In total, 265 discs were divided into 32 groups (n = 8/group) according to: (1) Material: *X*-tra fil and Admira Fusion *X*-tra; (2) Surface treatment: oxygen inhibition; matrix; Futurabond M+; Silane/Futurabond M+; Admira Bond; Silane/Admira Bond; ceramic repair system; and Silane/Cimara bond; and (3) Storage time: 24 h and 6 months. Each disc received three micro-cylinders from the same material. Specimens were subjected to micro-shear bond strength testing either at 24 h or 6 months. Data were analyzed using ANOVA/Tukey's test/Student *t*-test ($p = 0.05$). All experimental factors had significant effect on bond strength ($p < 0.0001$). Drop in bond strength was noticed in both materials after six months ($p < 0.05$), except for Admira Fusion *X*-tra treated with silane/cimara adhesive ($p = 0.860$). Both materials showed insignificant values with Admira bond either at 24 h or 6 months ($p = 0.275$ and $p = 0.060$, respectively). For other treatments, *X*-tra fil showed significantly higher values at 24 h and 6 months ($p < 0.05$). Ceramic repair system can be used to immediately repair both methacrylate- and ormocer-based composites.

**Keywords:** bulk fill; immediate repair; bond strength

## 1. Introduction

Nowadays, due to the abandonment of amalgam use, direct resin composites have become one of the most widely used restorative materials to restore damaged tooth structures [1]. Many efforts have constantly been deployed to improve resin composite material quality in order to increase the longevity of such restorations. Researchers and manufacturer developers seek not only to improve the quality of the material but also to achieve an acceptable clinical time. In this regard, simplified adhesives [2], flowable [3] and regular viscosity bulk fill resin composites [4] have been launched in the market to decrease the clinical procedural steps and accordingly shorten the restoration time. Flowable and regular viscosity bulk fill resin composites can frequently be used to restore class II cavities, as they showed equivalent marginal quality compared to conventional ones [5].

Restoration defects can be detected immediately in the form of sub-margins, under contours or in the presence of voids [6] that would necessitate instant repair, which could be deemed a "minimally

invasive approach" [7] to correct such defects. Studies on adhesion should not be limited to restorative material/hard tissue interface, but they should also be extended to evaluate the adhesion between restorative materials [8], as defects are not always at the restoration/tooth interface—they could also be confined within the restoration itself.

Different surface treatments, mechanical, physical, chemical or a combination of all were evaluated extensively in previous studies [9–21]. Among the techniques proposed to improve the repair bond strength of composite resins, mechanical roughening with diamond burs was recommended to eliminate the superficial layer and create irregularities in order to increase the bonding surface area [11,15,16,21]. Since the old composite no longer has its oxygen inhibited layer, the use of a silane primer as an intermediary agent enhances the wetting of fresh resin composites to aged composite substrates, promotes chemical bonding of resin to filler particles and increases the flow of low-viscosity adhesives on irregular surfaces [9,17,19,21]. Additionally, the application of bonding agents could be responsible for chemical bonding to organic matrix of the repaired resin composite [11,13,14,19]. Universal adhesives were recently introduced to enable bonding to various substrates, such as dental hard tissues, ceramics, resins and metals, without the need for separate primers [13,21]. However, it was reported that there was no universal surface treatment to repair defective resin composite materials, as it was shown to be material dependent [22]. Furthermore, the variously proposed repair protocols were based on studies with "low level of evidence" [23].

Recently, several studies [24–28] focused on the repair of methacrylate-based bulk fill resin composites, either with conventional resin composites or with the same bulk fill materials. They concluded that methacrylate-based bulk fill resin composites could be efficiently repaired with both materials. On the other hand, and to the best of our knowledge, the repair potential of ormocer-based bulk fill resin composite did not gain much attention in former studies.

The aim of this study was to evaluate the effect of different surface treatments and storage time on immediate bond strength durability of methacrylate- and ormocer-based bulk fill resin composites. The null hypothesis tested was that neither surface treatments nor storage time has any effect on immediate bond strength durability of methacrylate- and ormocer-based bulk fill resin composites.

## 2. Materials and Methods

Two bulk fill resin composites—a methacrylate-based (X-tra fil) and an ormocer-based (Admira Fusion X-tra)—were used in this study. An ormocer-based etch-and-rinse adhesive (Admira Bond), a universal adhesive (Futurabond M+) and ceramic repair system (Cimara) were also used as intermediate agents before repair procedures. The material (description), compositions (batch #) and manufacturer are listed in Table 1.

**Table 1.** Materials (description), composition (batch #) and manufacturer.

| Material (Description) | Composition (Batch #) | Manufacturer |
|---|---|---|
| *X*-tra fil (micro-hybrid bulk-fill resin composite) | Bis-GMA, UDMA, TEGDMA, silicate glass. Filler content: 86 wt% (1533445) | VOCO GmbH, Cuxhaven, Germany |
| Admira Fusion x-tra (ormocer-based bulk-fill resin composite) | Ormocer resin, silicon oxide fillers and glass fillers. Filler content: 84 wt% (1509036) | VOCO GmbH |
| Futurabond M+ (universal adhesive) | Dimethacrylates, fumed silica, acid-modified methacrylates, CQ, BHT, amine, ethanol, water (1428143) | VOCO GmbH |
| Admira Bond (ormocer-based etch-and-rinse adhesive) | Acid-etching gel: 34.5% phosphoric acid-etching gel (1411479) Bond: ormocer resin, dimethacrylates, HEMA, NaF, acid-modified methacrylates, CQ, BHT, acetone (1421529) | VOCO GmbH |

**Table 1.** *Cont.*

| Material (Description) | Composition (Batch #) | Manufacturer |
|---|---|---|
| Cimara (ceramic repair system) | SiC bur: SiC grinding low-speed bur<br>Adhesive: dimethacrylates, carbon acid-modified dimethacrylates, CQ, BHT, Amine, acetone (1414216)<br>Silane: Reactive silane, isopropanol, acetone, amine (1415052) | VOCO GmbH |

Bis-GMA: Bis-phenol A glycidyl methacrylate; UDMA: urethane dimethacrylate; TEGDMA: Triethylene glycol dimethacrylate; CQ: camphorquinone; BHT: butylated hydroxytoluene.

## 2.1. Study Design

A total of 256 resin composite discs (7 mm diameter and 2 mm thickness) were prepared and identified as substrate materials, onto which the repair procedures were performed. The discs were divided into 32 groups (n = 8 discs/group) according to the three main experimental factors in this study: factor 1: material, 2 groups (*X*-tra fil and Admira Fusion X-tra); factor 2: surface treatment, 8 groups (oxygen inhibition, matrix, Futurabond M+, Silane/Futurabond M+, Admira Bond, Silane/Admira Bond, ceramic repair system and Silane/Cimara bond); and factor 3: storage time, 2 groups (24 h and 6 months).

## 2.2. Specimens Preparation and Surface Treatments

All resin composite discs were prepared using split Teflon mold with a central hole of 7 mm diameter and 2 mm thickness. The mold was placed on a piece of polyester strip (Stripmat, Polydentia, Mezzovico, Switzerland) over a glass slide. Resin composite material was packed inside the central hole of the mold until the material was slightly over filling the mold. Afterwards, the repair surface treatments were assigned as follows:

1. **Oxygen-inhibited layer (OIL, n = 32 discs):** In this group, a piece of Teflon tape was placed over the resin composite disc and gently pressed with a glass slide to extrude excess material and create a smooth flat surface. The glass slide and the Teflon tape were removed, and the material was photo-polymerized in contact with air for 10 s and 20 s (*X*-tra fil and Admira Fusion X-tra, respectively), following the manufacturer's instructions, using LED photo-polymerization unit (Elipar, 3M ESPE, Seefeld, Germany; light output: 1200 mW/cm$^2$). The discs were removed from the mold and no further surface treatments were performed. These discs were used to test the cohesive strength of each material (positive control).

2. **Matrix (M, n = 32 discs):** After the material was packed inside the mold, a piece of polyester strip was placed on the top of the resin composite and gently pressed with a glass slide as described for OIL. The glass slide was then removed and the material was photo-polymerized using the LED photo-polymerization unit for 10 s and 20 s (*X*-tra fil and Admira Fusion X-tra, respectively), following the manufacturer's instructions. The discs were removed from the mold and no further surface treatments were performed. These discs served as negative control group.

The remaining 192 discs (n = 96/each resin composite material) were light-polymerized against the polyester strip and removed from the mold. They were then wet-ground over #600 SiC papers for 10 s, rinsed for 5 s and air-dried for 5 s, before the application of the following surface treatments:

3. **Futurabond M+ adhesive (FBM+):** The adhesive was applied with a microbrush (Single Tim, VOCO) and rubbed over the surface for 20 s, air-dried for 5 s and photo-polymerization for 10 s according to the manufacturer's instructions.

4. **Silane/Futurabond M+ adhesive (S/FBM+):** Silane was applied with a microbrush and left to react for 2 min. The silane was not air-dried according to manufacturer's instructions. The adhesive was applied and rubbed for 20 s, air-dried for 5 s, and photo-polymerized for 10 s.

5. **Admira Bond (AB):** Each disc was etched using 34.5% phosphoric acid etchant gel for 15 s, rinsed for 20 s and air-dried for 10 s. Adhesive was applied to disc surface using a microbrush and left

undisturbed for 30 s. The adhesive was gently air-dried for 5 s and photo-polymerized for 10 s according to the manufacturer's instructions.

6. **Silane/Admira Bond (S/AB):** Silane was applied and left to react for 2 min and no air dryness was performed. Admira Bond was applied for 30 s, air-dried for 5 s and photo-polymerized for 10 s.

7. **Ceramic Repair System (CRS):** Each composite disc was conditioned using a silicon carbide (SiC) grinding bur supplied by the manufacturer (Cimara bur, VOCO), in one direction for 5 s [20]. The bur rotated at 10,000 rpm using a slow speed handpiece (Sirona, T2 Revo-R 40, Sirona Dental System, Bensheim, Germany) mounted on electrically-controlled motor. The conditioned surface was thoroughly cleaned using gentle compressed air. Silane was applied and left to react for 2 min. Cimara adhesive was applied using a micro-brush, distributed over the surface using a gentle stream of air. The adhesive was left undisturbed for 20 s and photo-polymerization for 20 s according to manufacturer's instructions.

8. **Silane/Cimara Bond (S/CB):** Silane was applied over the surface and left to react for 2 min, but no air dryness was performed. Adhesive was applied, gently distributed over the surface, left undisturbed for 20 s and photo-polymerized for 20 s.

*2.3. Application of Repair Material*

Each resin composite disc (substrate) received 3 micro-composite cylinders from the same material by the aid of 3 polyethylene tubes. Polyethylene tubes (2 mm external diameter, 1 mm internal diameter and 0.7 mm height) were placed over each composite disc and were filled with the repair composite. The tubes were covered with a polyester strip and gently pressed to extrude excess material. The tubes were then photo-polymerized for 10 s or 20 s according to bulk fill material applied (*X*-tra fil and Admira Fusion X-tra, respectively). Half of the discs with their attached micro-cylinders (64 discs/each material) were stored for 24 h to evaluate the immediate bond strength and the other half (64 discs/each material) was stored in distilled water for 6 months to evaluate the bond durability. Distilled water was changed weekly until the end of 6-month storage time. Specimen preparation and repair procedures are illustrated in Figure 1.

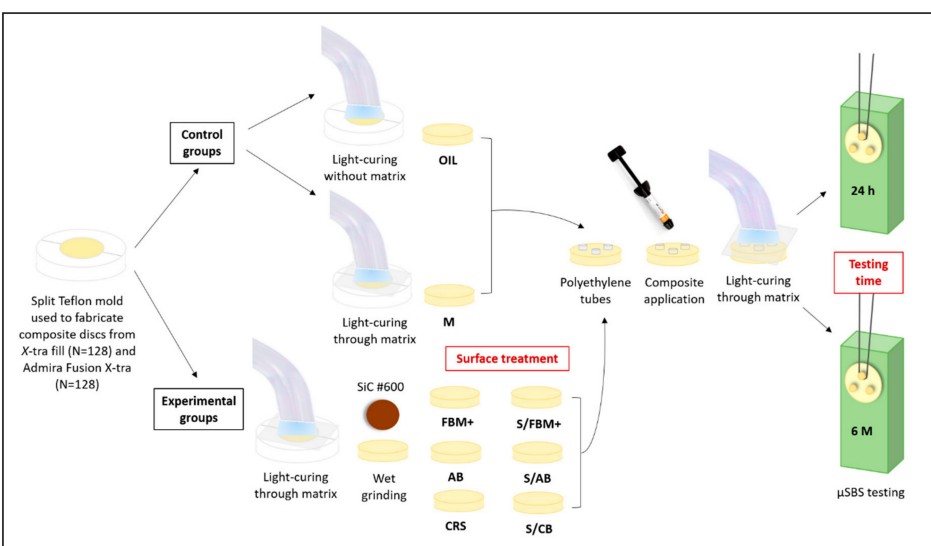

**Figure 1.** Specimen preparation and repair procedures.

*2.4. Micro-Shear Bond Strength (μSBS) Testing*

All polyethylene tubes were removed immediately prior to μSBS testing. Two vertical cuts were performed using a surgical blade to separate each tube into two halves, which were then carefully removed. Any excess adhesive resin or composite material around the micro-cylinders was also

carefully removed using the surgical blade. One pretest failed micro-cylinder specimen was detected only in Admira Fusion X-tra treated with S/AB at 24 h storage period. Micro-cylinders (n = 767 micro-cylinders) were then examined under magnification to verify absence of voids or defects at the bonding interface, in order to exclude defective micro-cylinders from testing. All examined micro-cylinders revealed no voids or defects at the interface and were subjected to micro-shear bond strength test.

Each composite disc with its bonded micro-cylinders was fixed onto a rectangular-shaped acrylic block (10 mm × 10 mm × 70 mm) using a cyanoacrylate adhesive [20]. The acrylic block was attached to the lower jig of a universal testing machine (Lloyd instruments LR5, Fareham, UK). An orthodontic wire of 0.2 mm diameter was carefully wrapped around each bonded micro-cylinder as close as possible to composite-composite interface and aligned parallel to applied shear force. The test was run at a crosshead speed of 0.5 mm/min until failure. The μSBS was calculated by dividing the load at debonding (Newton) by the bonded surface area (mm$^2$).

*2.5. Statistical Analysis*

The means of the three micro-cylinders from each disc were averaged and the overall mean of the group was calculated from eight discs. Statistical analysis was performed using SPSS statistical analysis for Windows (Version 20). Data were presented as the mean, standard deviation and percentage. Significance level was set at $p$ = 0.05. Three-way ANOVA was conducted to evaluate the effect of "material", "surface treatment", "storage time" and their interactions on μSBS. One-way ANOVA/Tukey's honestly significant difference test (HSD) post-hoc test were used for pairwise comparisons. Independent student t-test was used to compare different surface treatments and storage time between the two materials.

## 3. Results

Three-way ANOVA showed that all experimental factors and their interactions had a significant effect on the μSBS ($p$ < 0.0001).

Concerning X-tra fil (Table 2), light curing against the polyester strip revealed the highest μSBS after 24 h-storage, but was statistically insignificant with FBM+, S/FBM+, CRS and S/CB ($p$ > 0.05). Meanwhile, AB showed the lowest μSBS, which was not statistically significant with S/AB ($p$ > 0.05). On the other hand, after 6-month storage time, S/FBM+ exhibited the highest μSBS, which was statistically similar to FBM+, CRS and S/CB ($p$ > 0.05). When the material was treated with AB, it yielded the lowest μSBS value, which was statistically insignificant with M ($p$ > 0.05). In all surface treatments, μSBS results significantly dropped when tested after 6 months of storage compared to 24 h testing time ($p$ < 0.05).

**Table 2.** Mean ± standard deviation of μSBS in MPa for the effect surface treatment within each storage time, and the effect of storage time within each surface treatment; and percentage drop in μSBS of *X*-tra fil resin composite.

|  | 24 h | 6 Months | Percentage Drop | *p*-Value |
|---|---|---|---|---|
| **OIL** | 21.5 ± 4.8 [bcA] | 16.2 ± 1.9 [bB] | 24.65% | 0.011 |
| **M** | 28.3 ± 4.3 [aA] | 10.9 ± 2.8 [cB] | 61.48% | <0.001 |
| **FBM+** | 25.1 ± 3.6 [abcA] | 17.3 ± 2.1 [abB] | 31.07% | <0.001 |
| **S/FBM+** | 27.4 ± 5.0 [abA] | 21.2 ± 2.9 [aB] | 22.63% | 0.009 |
| **AB** | 14.1± 2.2 [dA] | 10.4 ± 2.2 [cB] | 26.24% | 0.004 |
| **S/AB** | 19.3 ± 4.1 [cdA] | 15.3 ± 2.1 [bB] | 20.73% | 0.035 |
| **CRS** | 26.3 ± 5.1 [abA] | 20.5 ± 3.5 [aB] | 22.05% | 0.020 |
| **S/CB** | 25.2 ± 3.0 [abcA] | 18.6 ± 2.6 [abB] | 26.19% | <0.001 |

Means with same superscript lowercase letters within each column and same superscript uppercase letters within each row are not statistically significant at $p$ = 0.05.

Regarding Admira Fusion X-tra (Table 3), there was no statistically significant difference at 24 h between S/FBM+, CRS, AB, S/CB and OIL, which yielded the highest μSBS values ($p > 0.05$). However, S/AB and FBM+ showed the lowest μSBS values after 6 months and were statistically insignificant ($p > 0.05$). In all surface treatments, there was a statistically significant difference between 24 h and 6 months ($p < 0.05$), except for S/CB, which showed no significant difference between both storage times ($p = 0.860$).

**Table 3.** Mean ± standard deviation of μSBS in MPa for the effect surface treatment within each storage time, and the effect of storage time within each surface treatment; and percentage drop in μSBS of Admira Fusion X-tra resin composite.

| | 24 h | 6 Months | Percentage Drop | *p*-Value |
|---|---|---|---|---|
| **OIL** | 12.9 ± 2.1 abcA | 9.6 ± 1.4 bcB | 25.6% | 0.003 |
| **M** | 11.9 ± 1.8 bcA | 7.5 ± 1.3 cB | 37% | <0.0001 |
| **FBM+** | 11.6 ± 2.7 cA | 7.9 ±1.1 cB | 31.9% | 0.005 |
| **S/FBM+** | 17.0 ± 3.7 aA | 8.4 ± 0.7 cB | 50.6% | <0.0001 |
| **AB** | 15.8 ± 3.3 abA | 8.6 ± 1.2 cB | 45.6% | <0.0001 |
| **S/AB** | 11.6 ± 2.0 cA | 8.1 ± 1.4 cB | 30.2% | 0.002 |
| **CRS** | 16.7 ± 2.3 aA | 11.1 ± 2.5 abB | 33.5% | <0.0001 |
| **S/CB** | 12.9 ± 2.4 abcA | 12.7 ± 1.2 aA | 1.6% | 0.860 |

Means with same superscript lowercase letters within each column and same superscript uppercase letters within each row are not statistically significant at $p = 0.05$.

For the effect of material type (Table 4), there was a statistically significant difference between the two materials ($p < 0.05$) at 24 h and 6-month storage times. Only AB surface treatment produced no statistically significant difference between the two materials either at 24 h or 6 months ($p = 0.275$ and $p = 0.06$, respectively).

**Table 4.** Mean ± standard deviation for the effect of material within each surface treatment and storage time on μSBS.

| | | X-Tra Fil | Admira Fusion X-Tra | *p*-Value |
|---|---|---|---|---|
| **24 h** | **OIL** | 21.5 ± 4.8 A | 12.9 ± 2.1 B | 0.001 |
| | **M** | 28.3 ± 4.3 A | 11.9 ± 1.8 B | <0.0001 |
| | **FBM+** | 25.1 ± 3.6 A | 11.6 ± 2.7 B | <0.0001 |
| | **S/FBM+** | 27.4 ± 5.0 A | 17.0 ± 3.7 B | 0.001 |
| | **AB** | 14.1± 2.2 A | 15.8 ± 3.3 A | 0.275 |
| | **S/AB** | 19.3 ± 4.1 A | 11.6 ± 2.0 B | 0.001 |
| | **CRS** | 26.3 ± 5.1 A | 16.7 ± 2.3 B | 0.001 |
| | **S/CB** | 25.2 ± 3.0 A | 12.9 ± 2.4 B | <0.001 |
| **6 months** | **OIL** | 16.2 ± 1.9 A | 9.6 ± 1.4 B | <0.001 |
| | **M** | 10.9 ± 2.8 A | 7.5 ± 1.3 B | 0.012 |
| | **FBM+** | 17.3 ± 2.1 A | 7.9 ±1.1 B | <0.0001 |
| | **S/FBM+** | 21.2 ± 2.9 A | 8.4 ± 0.7 B | <0.0001 |
| | **AB** | 10.4 ± 2.2 A | 8.6 ± 1.2 A | 0.060 |
| | **S/AB** | 15.3 ± 2.1 A | 8.1 ± 1.4 B | <0.0001 |
| | **CRS** | 20.5 ± 3.5 A | 11.1 ± 2.5 B | <0.0001 |
| | **S/CB** | 18.6 ± 2.6 A | 12.7 ± 1.2 B | <0.0001 |

Means with same superscript uppercase letters within each row are not statistically significant at $p = 0.05$.

## 4. Discussion

As replacement of failed restorations accounted for more than half of restorations placed [29], the decision to replace failed restorations often necessitates the removal of sound tooth structure; with subsequent increase in the size of prepared cavity and weakening of the remaining tooth

structures [30]. Term "repair" is considered when adding material is decided to repair the defect either without additional preparation or with minimum preparation including tooth and/or defective restoration [7].

Bulk fill resin composites were developed aiming to reduce the chair-side time wasted during the application of several increments as in conventional resin composites [31]. Admira Fusion X-tra is the bulk version of Admira Fusion. It is an "organically modified ceramic" that contains no classic monomers such as Bis-GMA, TEGDMA or HEMA to improve its "biocompatibility". Its chemical structure is based on silicon oxide innovative technology, not only in fillers, but also for the resin matrix (Admira Fusion and Admira Fusion X-tra, Technical Product Profile, VOCO GmbH).

From the results of this study, the null hypothesis must be rejected. All experimental factors ("material", "surface treatment" and "storage time") had significant effect of immediate repair μSBS. When *X*-tra fil was photo-polymerized against polyester strip, it yielded significantly higher repair bond strength compared to when the material was photo-polymerized in contact with air (Table 2). Although the presence of matrix prevents the formation of the OIL [31], the OIL did not guarantee a higher bond strength between layers of resin composites [32]. The degree of conversion of methacrylate-based bulk fill resin composites after 24 h exhibited significantly higher values in contrast to their immediate values [33]. This meant that there were still reactive monomer sites for attachment, to which the repair material could adhere successfully. In methacrylate-based resin composite, the high degree of polymerization could allow optimal adhesion between resin composite layers, regardless of the presence or absence of the oxygen-inhibited layer [32]. The presence of matrix provides a highly smooth flat repair surface that could help in the intimate adaptation of repair material over the substrate. The findings of this study were in agreement with that of a previous study [20] investigating the effect of the presence of the oxygen-inhibition layer on bond strength of methacrylate-based resin composites. On the other hand, after six months of storage (Table 2), a significant reduction in bond strength was detected when the material was light cured against the polyester strip, compared to the presence of the oxygen inhibition layer. It could be hypothesized that the low height (0.7 mm) of the composite micro-cylinders had allowed the passage of light during photo-polymerization to reach the oxygen inhibition layer on the substrate surface. Additional photo-polymerization might have played a role in enhancing the formation of covalent and secondary bonds and creation of an "interpenetrating network", which improved the "mechanical interlocking" [34] between the substrate layer and the overlying repair material. On the contrary, in case of polyester strip, the surface was not covered with an oxygen inhibition layer, accordingly further improvement in the adhesion through development of covalent bonds between the two layers (substrate and repair material) might not be expected, leading to a significant reduction in bond strength.

On the other hand, there was no statistically significant difference in μSBS values between oxygen inhibition and matrix application for Admira Fusion X-tra (Table 3). Although ormocer resin do not contain a traditional Bis-GMA monomer, its manufacturer (VOCO GmbH) that it exhibits an oxygen inhibition layer when the material is photo-polymerized in air, similar to methacrylate-based resin composite. It was reported previously that the presence of oxygen inhibition layer was not essential to improve the immediate μSBS of ormocer-based resin composite [21], though the latter was conducted on Admira Fusion resin composite, which is considered as the non-bulk version of Admira Fusion X-tra. In addition, the results of six-month storage showed also insignificant results between oxygen inhibition and polyester strip. This might prove that the presence of an oxygen inhibition layer was not essential to bond the different layers of the ormocer-based resin composite.

Both resin composites in this study could be repaired successfully using the ceramic repair system (CRS), either with or without the use of the SiC grinding bur. Previous studies showed that the use of ceramic repair system to repair immediate [21] and aged [21–35] resin composites improved the repair bond strength. According to manufacturer's instructions, SiC bur is used to condition the ceramic surface following preparation with abrasive bur. In case of resin composite, the step of using bur could be omitted. The SiC bur produced smooth smear layer surface deposits that dimmed the

grinding grooves on the surface that were already formed by #600 SiC paper [21]. When silane was applied, it was left to react for 2 min. This relatively long clinical time could allow the silane to coat exposed fillers and interpenetrate the micro-roughened surface created by grinding procedures. Silane was also not air dried according to manufacturer's instructions, which kept the silane layer undisturbed. When the Cimara adhesive applied to the pre-silanized surface, it was just distributed over the surface using a faint stream of compressed air (Ceramic repair system, instructions for use, VOCO GmbH). This might have resulted in formation of relatively thick adhesive layer that could act as stress absorbing layer during stresses developed during testing [21].

In this study, the use of intermediate bonding agents showed contradictory results between the two bulk fill materials. When tested after 24 h, universal adhesive (FBM+) showed repair bond strength values close to those of the unconditioned surfaces (control) within both resin composite materials. This could be attributed to: (1) the intimate adaptation of the low viscosity resin over the ground resin composite surface [36]; (2) the presence of phosphate groups in FBM+ could contribute to the efficient wetting and bonding to the inorganic fillers in resin composite [37]; (3) for ground resin composite surfaces, the use of intermediate adhesive layer could compensate for the loss of unreacted methacrylate groups and render the newly added resin material less viscous and bond to the exposed fillers on the ground surface [38]; (4) the presence of grinding grooves on the surface could act as micro-retentive sites, into which adhesive, especially in self-etching mode, could be interlocked [21]; (5) the hydrophilic nature of the universal adhesive might offer an optimum wetting of the substrate that could promote the mechanical interlocking through the penetration of the adhesive into the created grinding grooves [19]. The results of this study came in accordance with the results of previous studies [24–26,28], though they evaluated the effect of the use of universal adhesives on repair bond strength of aged bulk fill resin composites. Hence, it appears that universal adhesives could efficiently repair bulk fill resin composites.

On the contrary, a significant drop in the repair bond strength was reported when Admira Bond was used to repair *X*-tra fil resin composite. However, this was not the case with Admira Fusion X-tra, as Admira Bond results showed significant improvement in the repair bond strength compared to the unconditioned control group using the matrix. In spite of the significantly low initial degree of conversion for the methacrylate-based resin composite [33], the difference in resin composition between the adhesive applied and resin composite might have contributed to these results. Admira Bond is an ormocer-based adhesive, hence the chemical adhesion through the monomer chains entanglement between the ormocer resin of adhesive and that of Admira Fusion X-tra resin composite might be accepted. This was not the condition when methacrylate-based resin composite was repaired with ormocer-based resin adhesive. This might raise the importance of resin matrix compatibility between the adhesive and resin composite in terms of immediate repair potential of the newly introduced bulk fill resin composites. Surprisingly, it was formerly reported that Admira Bond did not improve the immediate repair bond strength of the non-bulk version of the ormocer-based resin composite (Admira Fusion) compared to unconditioned surface (matrix) [21]. This could be ascribed to the low degree of conversion [33] of Admira Fusion X-tra bulk fill resin composite, with low transformation from gel phase to glass phase during photo-polymerization [39]. This unique property of the bulk fill resin composites was not observed in the non-bulk fill (conventional) resin composites, as the latter showed no significant difference in the degree of conversion initially or after 24 h [33] and exhibited a fast transformation from gel phase to glass phase during photo-polymerization [40].

The use of silane in this study prior to the application of the universal adhesive did not improve the repair bond strength of *X*-tra fill, but it had a significant positive influence on the repair bond strength of Admira Fusion X-tra compared to universal adhesive without silane pre-treatment. The difference in filler size between both resin composites might be the reason of these findings. *X*-tra fil is a micro-hybrid resin composite with its fillers reaching up to 10 μm in size [40]. On the other hand, Admira Fusion X-tra is a nano-hybrid resin composite with fillers size ranging from 0.1 μm up to 5 μm [39]. The smaller the filler size, the higher the surface area available for bonding, which might

enhance the interaction of silane with the exposed fillers. Subsequently, this could also improve the adhesion of the universal adhesive to resin composite surface. The proposed explanation still warrants further investigation.

The bond strength results of Admira Fusion in this study were in disagreement with the results of El-Askary et al. [19,21] It was reported that silane enhanced the immediate repair bond strength of nano-hybrid resin composite, when its surface was wet rather than dry condition [19]. In the presence of water, silane could efficiently react with the glass filler's hydroxyl groups to form highly cross-linked siloxane groups [41], which might lead to improvement in the chemical bond between substrate and repair composites via the intermediate adhesive layer [21]. Although the surface of Admira Fusion was repaired in dry condition in the study of El-Askary et al. [21] as in the present study, the application of silane yielded different results. The lower immediate degree of conversion of Admira Fusion X-tra [33] results in slow transformation rate from gel phase to glass phase during photo-polymerization with slow rate of polymerization initiation [39]. This might be the reason behind the presence of monomer sites of attachment just after photo-polymerization; that could lead to strong bonding between silane and resin matrix, resulting in significantly higher bond strength.

The results of methacrylate-based (*X*-tra fil) composite came in disagreement with the results of Aquino et al. [42]. They reported that silane pretreatment of bur treated methacrylate-based bulk fill resin composite enhanced the repair bond strength, and the use of abrasive bur/silane surface treatment could be an alternative treatment in situations where intraoral sandblasting is not available. The difference in the tested bulk fill resin composites might be the reason of such disagreement.

Interestingly, silane showed a significant reduction in repair bond strength of Admira Fusion X-tra when applied before Admira bond. However, it showed a numerical increase in bond strength of *X*-tra fil. According to manufacturer's instructions, Admira Bond was applied over the surface and left undisturbed for 30 s. Admira Bond contains acetone as solvent, which has a high vapor pressure and evaporation rate [43]. Furthermore, air dryness of Admira bond was performed gently for 5 s. Leaving the adhesive layer for 30 s could be considered a sufficient time to permit the evaporation of the highly volatile acetone from the adhesive layer. This might have created a thick adhesive layer over the silane layer. The presence of thick multilayer formed from silane and adhesive might create a multiphase thick layer, resulting in a weak bond [44]. It could be also assumed that the application of an intermediate layer (silane) had restricted the direct chemical interaction between Admira Bond and Admira Fusion X-tra. On the other hand, *X*-tra fil behaved differently with silane application prior to Admira Bond. Silane acted presumably as an intermediary that enhanced bonding between the two materials having dissimilar resin matrices, as stated previously.

*X*-tra fil resin composite showed higher immediate repair bond strength compared to Admira Fusion X-tra, either at 24 h or 6-month storage periods, except when Admira Bond was used. It was previously reported that shear bond strength between dental materials depended on the substrate material strength, onto which the other material was applied [45]. It was shown that *X*-tra fil had superior mechanical surface properties compared to Admira Fusion X-tra [46,47]. As tensile stresses generated near the application load point in shear test using the wire loop [48,49], the strength of the material to resist such stresses could be of importance. This could interpret why the repair bond strength of *X*-tra fil was significantly higher than that of Admira Fusion X-tra. Admira Bond is an acetone containing adhesive with 30 s application time (Admira Bond technical product profile, VOCO GmbH). This relatively long clinical application time might result in a rapid evaporation of acetone, which has a high vapor pressure [43], and consequently a thick viscous adhesive layer [21] might have been formed on the surface of substrate resin composites. This thick layer might act as stress absorbing layer that resisted the tensile stresses generated during testing, leading to insignificant repair bond strength results between the tested bulk fill resin composites.

Immediate repair bond strength dropped significantly after six months of storage, even with the use of silane prior to the adhesive application, except for Admira Fusion X-tra when treated with silane/Cimara adhesive. The hydrophilic nature of the adhesives used in this study could encourage the

absorption of water [50,51], which could lead to a degradation of the adhesive layer during storage [19]. Furthermore, silane bond could be degraded in the presence of water, which could be due to its hydrolytic instability [41]. The use of silane before the application of Cimara adhesive preserved the repair bond strength of Admira Fusion X-tra after six months of storage. It could be hypothesized that the innovation in the matrix technology of ormocer-based resin composite (Admira Fusion X-tra, technical product profile, VOCO GmbH), besides the long clinical application time of both silane (2 min) and Cimara adhesive (20 s) might provide an adhesive layer capable to resist the degradation during storage. This hypothesis may need further investigation.

It was obvious that the percentage drop in bond strength was lower in Admira Fusion X-tra (1.6–50.6%) compared to X-tra fil (20.73–61.48%). This could encourage further research with longer storage time to detect if further deterioration in the repair bond strength of ormocer-based resin composite would occur. The results of this study confirmed that "there is no one optimum repair technique that fits for all composites [22]". It seemed that the universal adhesive used in this study can optimally repair methacrylate-based resin composites; while the ormocer-based adhesive is preferably used to repair ormocer-based resin composite. The application of silane is not necessary in immediate repair of methacrylate-based resin composite, but it enhanced the repair bond strength of ormocer-based adhesive to repair ormocer-based resin composite.

Clinicians believe that immediate repair of resin composites is as simple as the application of resin composites in the layering technique. This concept should be reconsidered as the success of immediate repair of resin composite differs from one type to another and from an adhesion protocol to another. As teaching of repair of failed restorations is growing faster in most of dental schools [52–54], this should inspire us, as researchers, to exert more efforts and give more attention to find out a common, simple, economic and yet unsophisticated repair protocols for the different resin composites available in the market.

## 5. Conclusions

From the results of this study, the following could be concluded:

1.  Immediate repair bond strength of methacrylate-based resin composite was superior to ormocer-based composite.
2.  Ceramic repair system can successfully repair both methacrylate- and ormocer-based resin composites.
3.  The use of universal adhesive with methacrylate-based composite and ormocer-based adhesive with ormocer-based composite showed promising results.
4.  Silane application before universal adhesive, improved the immediate repair bond strength of ormocer-based composite, with no effect on methacrylate-based one.
5.  Adhesive/resin composite matrix compatibility should be considered during immediate repair of resin composites with different resin matrix.
6.  None of the bonding protocols succeeded to maintain the repair bond strength after six months of storage, except for ormocer-based composite treated with silane/Cimara adhesive.

**Author Contributions:** Conceptualization, F.S.E.-A. and M.Ö.; methodology, F.S.E.-A. and S.A.B.; validation, F.S.E.-A., S.A.B. and M.Ö.; statistical analysis, F.S.E.-A.; resources, F.S.E.-A., S.A.B. and M.Ö.; data curation, F.S.E.-A. and S.A.B.; writing—original draft preparation, F.S.E.-A. and S.A.B.; writing—review and editing, F.S.E.-A., S.A.B. and M.Ö.; visualization, F.S.E.-A., S.A.B. and M.Ö.; supervision, F.S.E.-A. and M.Ö. All authors have read and agreed to the published version of the manuscript.

**Funding:** This research received no external funding.

**Acknowledgments:** The authors would like to thank VOCO GmbH, Cuxhaven, Germany for supplying the materials tested in this study.

**Conflicts of Interest:** The authors declare no conflict of interest.

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
