# Peer review of "Effect of Surface Treatment and Storage Time on Immediate Repair Bond Strength Durability of Methacrylate- and Ormocer-Based Bulk Fill Resin Composites"

_applsci, doi:10.3390/app10228308_

Round 1

Reviewer 1 Report

In general, the paper is well written, but some main methodology procedures are missed. At the end of this study, we did not receive main outcome  and message for clinical use. Methodology is not clear at all. Some improvements are suggested. They all are provided in more details below.

  • Abstract of the manuscript should be supplemented with the brief information concerning the novelty of the studies and their meaningless.
  • Additional paragraph concerning the existing repair protocols  needs to be added to the Introduction section (one sentences concerning this issue have been added in line 52), but this subject should be more widely described. Next, the same section should be supplemented with brief information on the procedures currently used for the repair of the restoration defects.
  • Section 2.1. There are few questions about laboratory procedures- It is not clear why did you choose intervals of  24h and 6 months for storage time?It was not clear what do you want to mimic? And what about temperature - it is not described? Are specimens thermocycled and if not why? How did you simulate aging of materials?
  • At the end , characterize briefly the mentioned surface treatments as well as their impact on the bond strength of tested materials and give the highlights of the research.

Reviewer 2 Report

Dear authors:

Congratulations for the research done.

I have some comments that follow:

  • I understand that manufacturer recommend 10 seconds for X-tra fil composite, but they recommend 20 seconds for Admira Fusion. It would be safer to standardize 20 seconds for both materials? Is any chance of having influence of this lack of curing as seen in the results of OIL and M groups right 24hs and 6 months after?
  • No need for tables 2, 3 and 4. Significant interaction between groups explained in other tables and in the text is enough.
  • Table 5. Use uppercase letters shows statistical differences between 24 hs and 6 months.
  • The same in table 6.
  • Use letters to show statistical differences between groups in table 7 as well.
  • LINES 351 TO 258: It would be fundamental for you test to check SEM images of samples. In special when discussing failure modes, thickness, influence of surface treatment. This is a piece missing in this test and is mandatory in an adhesive assessment. Without images is difficult to make assumptions like in this paragraph.
  • LINE 272 again should be analyzed by images.
  • LINE 315. “The smaller the filler size, the higher the surface area available for bonding”. Please relate this sentence justifying the effect of silane in your results.
  • IN LINE 315: “ The smaller the filler size… “ Authors should state an explanation on silane X size of fillers. What is the reason and explanations. Again, SEM of materials used could help discussion.
  • As this paper was submitted probably in the end of 2019 or beginning of 2020, some papers could be added like: RepairBond Strength and Leakage of Non-Aged and Aged Bulk-fill Composite.Aquino C, Mathias C, Barreto SC, Cavalcanti AN, Marchi GM, Mathias P.Oral Health Prev Dent. 2020 
